# A computational approach to fighting type 1 diabetes by targeting 2C Coxsackie B virus protein with flavonoids

Shahid Ullah[1]☯*, Zilong Zheng[2]☯, Wajeeha Rahman[1], Farhan Ullah[1], Anees Ullah[1], Muhammad Nasir Iqbal[3], Naveed Iqbal[3], Tianshun Gao[2]*

1 S Khan Lab Mardan, Khyber Pakhtunkhwa, Pakistan, 2 Big Data Center, The Seventh Affiliated Hospital of Sun Yat-sen University, Shenzhen, P. R. China, 3 Department of Bioinformatics, Institute of Biochemistry, Biotechnology and Bioinformatics, The Islamia University of Bahawalpur, Bahawalpur, Pakistan

☯ These authors contributed equally to this work.
* drskbioch@gamil.com (SU); gaotsh3@mail.sysu.edu.cn (TG)

**Data Availability Statement:** All the data is available in the main paper and in supplementary tables, and will be freely available under journal rule.

## Abstract

Autoimmune diabetes, well-known as type 1 insulin-dependent diabetic mellitus (T1D). T1D is a prolonged condition marked by an inadequate supply of insulin. The lack is brought on by pancreatic cell death and results in hyperglycemia. The immune system, genetic predisposition, and environmental variables are just a few of the many elements that contribute significantly to the pathogenicity of T1D disease. In this study, we test flavonoids against Coxsackie virus protein to cope the type 1 diabetes. After protein target identification we perform molecular docking of flavonoids and selected target (1z8r). then performed the ADMET analysis and select the top compound the base of the docking score and the ADMET test analysis. Following that molecular dynamics simulation was performed up to 300 ns. Root means square deviation, root mean square fluctuation, secondary structure elements, and protein-ligand contacts were calculated as post-analysis of simulation. We further check the binding of the ligand with protein by performing MM-GBSA every 10 ns. Lead compound CID_5280445 was chosen as a possible medication based on analysis. The substance is non-toxic, meets the ADMET and BBB likeness requirements, and has the best interaction energy. This work will assist researchers in developing medicine and testing it as a treatment for Diabetes Mellitus Type 1 brought on by Coxsackie B4 viruses by giving them an understanding of chemicals against these viruses.

## Introduction

People with diabetes type 1 have abnormally high blood glucose levels because their systems are unable to manufacture insulin, a hormone that causes the condition. This occurs because the body strikes the tissues that produce insulin in the pancreas, inhibiting it from making insulin in the pancreas, preventing it from making the type of insulin at all. To keep alive, we all need insulin. It serves an important purpose. It allows blood sugar to enter our cells, supplying energy to our bodies. Even if a person develops type 1 diabetes, the body keeps converting

**Funding:** This research is supported by National Natural Science Foundation of China [32100434] and Research Start-up Fund of the Seventh Affiliated Hospital, Sun Yat-sen University [ZSQYBRJH0020]" Also The funders had the main role in study design, data collection and analysis, decision to publish, or preparation of the manuscript.

**Competing interests:** The authors have declared that no competing interests exist.

carbs from food and beverages into sugar. While glucose flow into the blood, however, there is insufficient insulin to allow it to get stored in the body [1,2]. T1D can manifest itself at any maturity, causing it one of the highly prevalent chronic illnesses in children. Adult-onset type 1 diabetes is more frequent than T1D in childhood and might be misdiagnosed as type 2 diabetes. T1D, which affects 5% to 10% of individuals with diabetes, has progressively grown in incidence and prevalence. A methodical study and extensive analysis found that the frequency of T1D was Fifteen per 100 thousand people, with a global prevalence of 9.5%. Furthermore, there is a large geographical variation in prevalence across the world. Finland and similar Northern European nations have the most reported cases, with rates around 400 times greater than China and Venezuela, which have the lowest known incidences [3–5].

Coxsackievirus B (CVB) is a member of the enterovirus kinds most likely to migrate from the mucous membrane of the intestines to the pancreas in the etiology of diabetes type 1 Mellitus. (T1DM). In the United States, Europe, Asia, Africa, and Australia, a couple of research studies comprising 4,448 and 5,921 individuals, each, verified the importance of the connection between the detection of infection with enteroviral evidence in various human tissue specimens and the likelihood of acquiring islet autoimmunity or T1DM [6,7]. In Type 1 diabetes, a postponed, progressive autoimmune development that may extend for years prior to the onset of apparent illness culminates in selective beta-cell dysfunction [8,9].

CVBs can infect human beta-cells since they are cytolytic viruses. The enterovirus family, which includes the coxsackievirus, is found in the human digestive system. Hepatitis A virus, polioviruses, and hand, foot, and mouth disease (HFMD) are all members of this enterovirus family. This virus transmits quickly from one person to another, generally by direct contact or contact with feces-contaminated surfaces. Because the virus can survive without a host for several days, propagating is very simple [10–12].

In addition, 66 separate serological investigations have identified human enterovirus serotypes. Enteroviruses come in four main varieties. These groups comprise Coxsackie A and B viruses, polioviruses, and echoviruses. The biological characteristics of these classes coincide while being distinct classes. There may be a link between Coxsackie viruses and type 1 diabetes. It has been believed that this theory exists for over 40 years. The nature of this relationship did not, however, support it. Animal models were employed to verify this connection [13]. This viral RNA has been found in the bloodstream of more than fifty percent of T1D affected at the moment of illness start, according to epidemiological data, which revealed a rise in the prevalence of the condition after enterovirus epidemics [14]. These models highlighted the association between T1D and enteroviruses.

A few of the Coxsackie virus B4 isolates from affected with type 1 diabetes with acute onset have been shown to produce diabetes in mice. There are several potential methods by which enteroviruses may initiate or quicken the degenerative processes most important to medical type 1 diabetes. Several enterovirus strains can reproduce in cultured human islet cells, inhibit insulin production, and, in rare instances, result in cell death [15,16].

Coxsackievirus B (CVB) serotypes may have a function in the etiology of type 1 diabetes, according to epidemiological research, although their precise involvement is yet unknown. We have anticipated a CVB1 drug from flavonoids in the current work. Natural remedies are gradually becoming more and more popular because they don't have any negative side effects in the treatment of brain disorders all over the world. These compounds play a large variety of roles in biological processes. Flavonoids, a class of low molecular weight phenolic compounds, are becoming more and more well-liked due to their wide range of health advantages, their ability to exert a variety of biological properties, including protection from neurological diseases, and their use in nutraceutical, pharmaceutical, medicinal, and cosmetic applications [17,18].

## Materials and methods

### Preparation of target protein structure and data for docking

Coxsackie B4virus has been detected in recent research employing PCR tests on a variety of diabetes patients. The family of picornaviruses includes coxsackie B4 viruses. These viruses are classified as small RNA viruses because they have a single positive component of RNA. Medicine should be used to address the Coxsackie B4 viral protein, which causes T1D. Coxsackie B4 virus, which kills pancreatic beta cells and causes T1D, is the target of the current in silico project, which aims to find active lead compounds against it using computational methods. The 3D configuration of the targeted protein was retrieved from RCSB PDB by using its specific PDB ID 1Z8R [19]. PDB, the online internet information portal provides access to 3D structural data of macromolecules (proteins, DNA, and RNA) [20]. MODELLER was used for loop refinement [21]. Swiss PDB Viewer [22] and RAMPAGE were used to optimize and minimize the protein crystal structure. RAMPAGE created a Ramachandran Plot that revealed no protein conflicts. The plot also shows which residues are in the favored, allowed, and outlier zones [23]. CASTp (Computed Atlas of Surface Topography of Proteins) predicted protein target binding sites. CASTp 3.0 identifies and quantifies protein topography reliably and thoroughly [24]. Docking was set up with a class of naturally occurring chemicals "flavonoids". Flavonoids are secondary polyphenolic chemicals found in plants and are a common component of human diets. Flavonoids consist of two phenyl rings and one heterocyclic ring, totaling 15 carbon atoms. Flavonoids' 2D structures were constructed and minimized. 37 flavonoids compounds were chosen from literature.

### Molecular docking

The top 37 compounds from flavonoids were chosen after sorting and screening. Using Auto-Dock Vina [25], energy dissipated while binding was measured, and protein-ligand interactions were evaluated, after docking these molecules with the receptor. PyMOL [26] was used to create complex receptor and ligand files, whereas BIOVIA Discovery Studio [27] was applied to find interactions in two dimensions.

### ADMET analysis

To establish the drug-likeness and toxicity characteristics of compounds, the pkCSM [28] and QikProp developed by Professor William L. Jorgensen [29] were utilized that are reported as essential and valuable tools for evaluation of important druglike descriptors like adsorption, dissemination, and breakdown, elimination, and toxicity (ADMET). These tools are also employed for predicting lead likeness concerning mutagenicity the carcinogenicity. Complete ADMET analysis results are uploaded in S1 Table.

### Lead identification

Researchers employed metrics like docking score, ligand-protein interactions, partial coefficient logP, rotatable bonds, rings, Polar Surface Area (PSA), Blood-Brain Barrier, and Ames Toxicity to narrow down the pool of potential inhibitors to a manageable set. 2C Coxsackie B virus protein inhibitor was selected based on their low binding affinities, strong lead-likeness scores, and positive interactions.

### MD simulation, PCA and DCCM

Schrödinger LLC's Desmond program was used for studying 300 ns MD simulations. [30]. The first crucial stage in molecular dynamics modeling, receptor-ligand docking, gives a fixed

image of a ligand's binding location at a protein's binding site [31]. By including Newton's classical calculation of action, MD simulations often forecast the status of lead compounds in a biological environment. [32,33].

The Protein Preparation Wizard in Maestro was used to perform preprocessing (optimization and minimization) on the receptor-ligand complex. In this process, steric conflicts, poor contacts, and deformed geometries were eliminated. All structures were built using the System Builder tool, and the OPLS_2005 force field was utilized with the solvent model TIP3P (Intermolecular Interaction Potential 3 Points Transferable), an orthorhombic box [34]. Throughout the simulation period, 310K temperature and 1atm pressure were utilized to imitate physiological circumstances while opposite ions were added to counteract the models, and 0.15M sodium chloride was injected. The models were made looser before the simulation. For examination, frames were saved after every 50 ps, and the binding of protein-ligand was determined over time using RMSD. Using the R package "Bio3D," the principal component analysis (PCA) and dynamic cross-correlation matrix (DCCM) were examined [35]. Simulation trajectory file could be found at this given link: https://drive.google.com/file/d/1hpx9TGYe2nA4z9VOhvwgrvsoBV7Y8UEM/view?usp=sharing

## Results and discussion

The 3D structure of the target protein (1Z8R) was obtained from Protein Data Bank. The total structure weight is 18.49 kDa. Fig 1 depicts the protein structure after loop refinement, optimization, and minimization. Ramachandran plot is shown in S1 Fig. The structure's overall quality was 98 percent, with highly preferred observations. In the plot, all other residues are displayed as circles, while glycine is plotted as triangles and proline as squares. The orange areas are the "favored" areas, the yellow areas are the "allowed" areas, and the white areas are the "disallowed" areas. AutoDock Vina performed the docking of the top hits. ADMET (absorption, distribution, metabolism, excretion, and toxicity) study was accomplished via QikProp and pkCSM. The top 10 compounds are included in Table 1 based on ADMET and docking findings.

In Table 1, mol_MW represents the molecular weight, which should be between 130.0 and 725.0, and donorHB is the projected amount of bonds that the solute would give to water molecules. accepted, the projected amount of bonds the solute would accept from water molecules in an aqueous solution can be a non-integer value with a recommended range of 0.0–6.0. This is because the value is an average across several different configurations. Given that values are calculated as an average over multiple states, they may not all be integers. It operates between 2.0 and 20.0. The Octanol/water partition coefficient, estimated to be in the range of -2.0 to 6.5, is denoted as QPlogPo/w. QPlogHER, Value of the inhibitory concentration (IC50) for the blockade of HERG K+ channels. Negative values below -5 are the cause for alarm. QPPCaco Caco-2 cell permeability prediction, expressed as several nanometers per second. The gut-blood barrier can be mimicked using Caco2 cells. The results of QikProp are for passive transport only. In the range of 0–25, consider it poor, and anything beyond 500 is excellent. QPlogBB Expected brain/blood separation ratio. Dopamine and serotonin, for instance, are CNS-negative because they are too polar to cross the blood-brain barrier, with predicted ranges of—3.0 to—1.2 when using QikProp to predict orally administered drugs. QPlogKhsa and Human serum albumin binding predictions range from -1.5 to 1.5.

Following lead identification, one compound CID: 5280445 was discovered as the most active of all compounds. Fig 2 depicts the best one's 2D interactions. The properties of the best one are shown in Table 1. Tetrahydroxyflavone luteolin has four hydroxyl groups at positions 3', 4', 5, and 7 on its molecular structure. It is believed to function as a key antioxidant, free

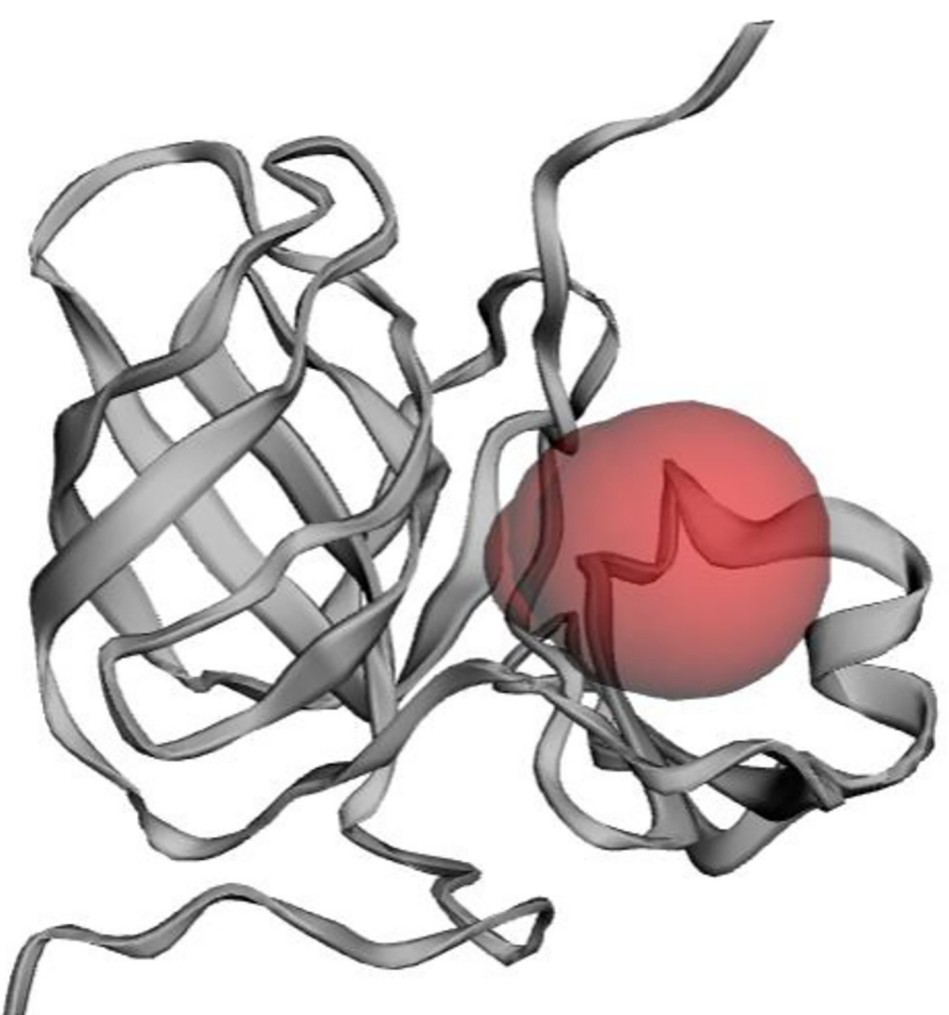

**Fig 1. 3D structure of protein retrieved from PDB after minimization showing binding pocket.**

radical hunter, anti-inflammatory, immunity modulator, and active against several malignancies in the human body [36]. The optimal chemical complex with the protein target was simulated using molecular dynamics simulation for 300 ns. Desmond's simulated trajectories were analyzed. Root-mean-square-deviation (RMSD) and root-mean-square-fluctuation (RMSF) values, as well as protein-ligand interactions, were determined using MD trajectory analysis.

Fig 3 shows the time-dependent variation in RMSD estimates for C-alpha particles in ligand-bound proteins. The RMSD plot shows that the complexed protein (PDB ID: 1z8r)

**Table 1. Table showing ADMET properties, binding affinity, and pharmacophore score of top compounds.**

| C_ID | mol_MW | donorHB | accept | QPlogPo/w | QPlogHERG | QPPCaco | QPlogBB | QPlogKhsa | Binding Affinity (Kcal/mol) |
|---|---|---|---|---|---|---|---|---|---|
| 5280445 | 286.24 | 3 | 4.5 | 0.941 | -5.023 | 45.027 | -1.91 | -0.205 | -6.6 |
| 65084 | 306.271 | 6 | 6.2 | -0.227 | -4.706 | 18.622 | -2.415 | -0.566 | -6.4 |
| 5281792 | 360.32 | 5 | 7 | 1.195 | -4.11 | 2.138 | -3.442 | -0.564 | -6.4 |
| 440735 | 288.256 | 3 | 4.75 | 0.876 | -4.633 | 50.253 | -1.798 | -0.217 | -6.3 |
| 445154 | 228.247 | 3 | 2.25 | 1.976 | -5.291 | 280.02 | -1.286 | -0.172 | -5.8 |

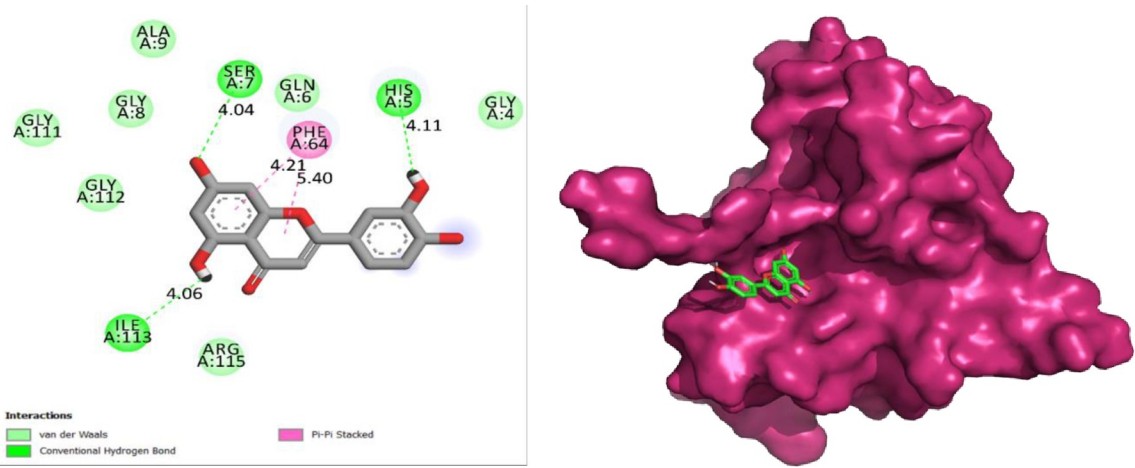

**Fig 2. Interactions of the lead compound with protein target showing interacting residues and length of bond (5280445_1z8r).**

stabilized at 20 ns. Once the simulation begins, the RMSD stays in the range of 0.5 Angstrom for the rest of the run, which is fine. The experiment validated the general belief that the building was sturdy. Throughout the simulation, there was no significant change in the Ligand Fit to Protein. The RMSD numbers would fluctuate suddenly, sometimes going up and sometimes down. After equilibrium was reached, there was no change in the ligand's RMSD.

Protein dynamics are characterized by PCA (Principal Component Analysis) [37]. Observing collective trajectory motions during MD simulations is a valuable tool. Graph of eigenvalues (protein) vs eigenvector index (eigenmode) for the initial 20 forms of action (NPA022882_1z8r) (Fig 4A). The eigenvalues depict hyperspace eigenvector fluctuations. In simulations, eigenvectors with higher eigenvalues regulate the proteins' total mobility. The top five eigenvectors in our systems showed dominant movements and had larger eigenvalues (35.4–72.9%) than the other eigenvectors, which had low eigenvalues. More than 50% of all changes were covered by the first three PCs (PC1, PC2, and PC3) that were plotted. According to the Fig 4A plots, PC1 clusters had the largest variability (35.43%), PC2 showed variability

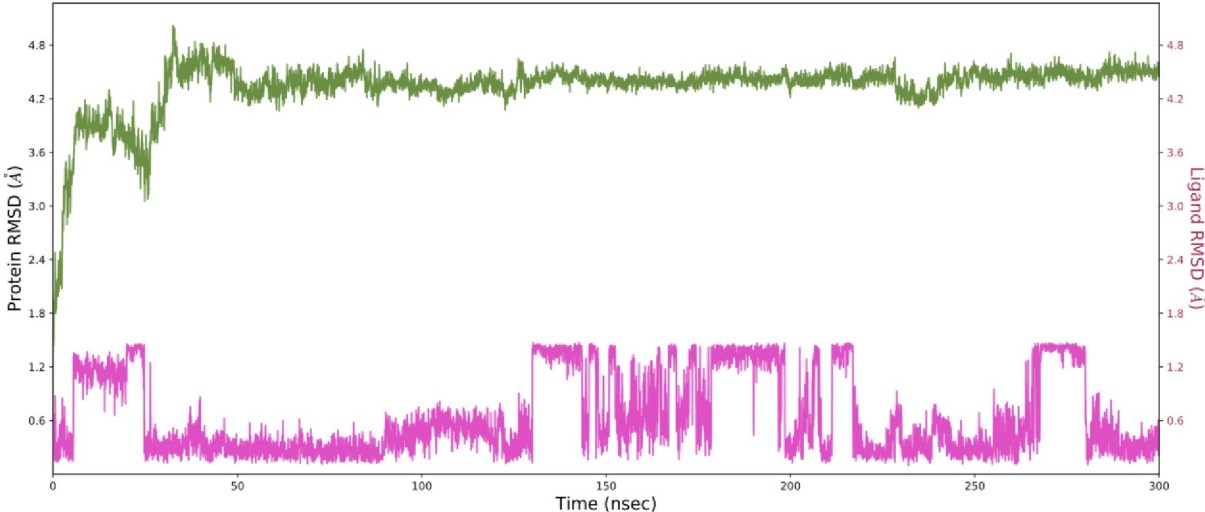

**Fig 3. Variation in the root mean square distance (RMSD) between the C-alpha atoms of proteins and lead over time (5280445_1z8r).**
The pink color shows the RMSD of the lead compound and green shows the RMSD of the protein target over time.

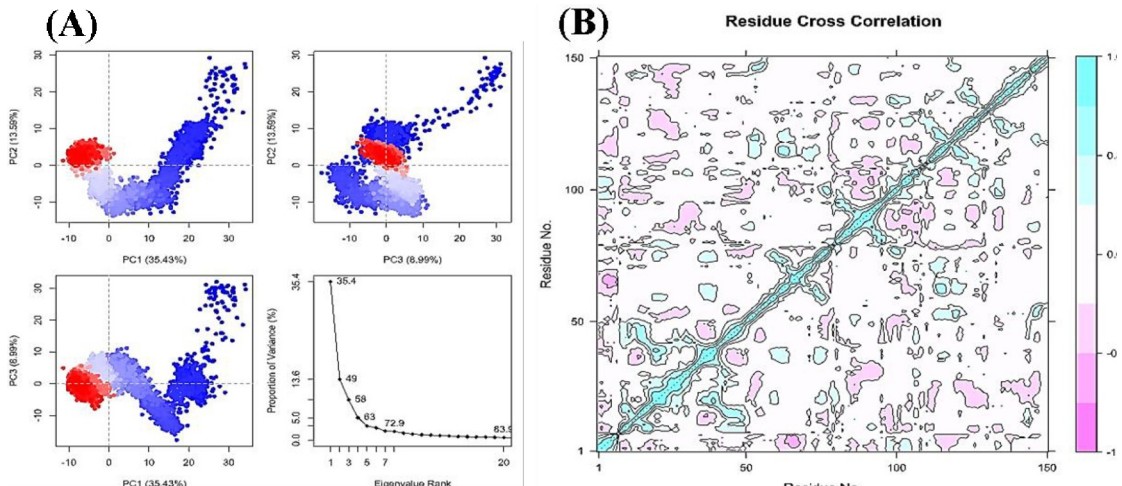

**Fig 4.** (A) Principal Component Analysis eigenvalue plotted versus the percentage of variance (NPA022882_1z8r). The varying areas are displayed in three separate sections. Variations in PC1, PC2, and PC3 add up to 35.43 percent, 13.59 percent, and 8.99 percent, respectively. (B) Complex 5280445_1z8r dynamic cross-correlation map. The residues' positive and negative correlations are depicted by cyan and purple, respectively.

(*13.69%*), and PC3 had the lowest variability (*8.99%*). As a result of its low variability, PC3 has a more compact structure than PC1 and PC2 and is thought to have a highly steadied protein-ligand binding. Straightforward grouping in the PC subspace showed conformational variations across all groups, with blue exhibiting the most significant mobility, white indicating intermediate movement, and red indicating less flexibility.

CID: 5280445 and the 1z8r protein were shown to be significantly correlated with one another, as seen by their high pairwise cross-correlation coefficient value on the cross-correlation map (Fig 4B). Magenta represents anti-correlated residues (-0.4), whereas cyan represents correlated residues (>0.8). It is clear from a large number of pairwise correlated residues between the 1z8r protein and ligand that their binding connection is stable.

Fig 5 illustrates the Residue-wise Root Mean Square Fluctuation (RMSF) of the ligand-coupled protein. Based on MD trajectories, we know that residues with higher peaks are in loop

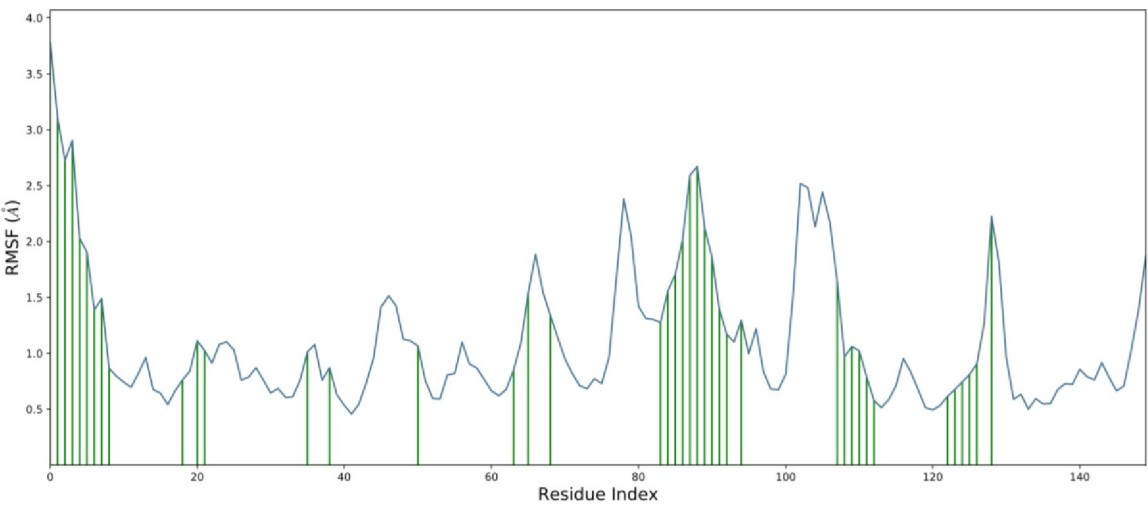

**Fig 5. RMSF of protein complexed with the ligand.**

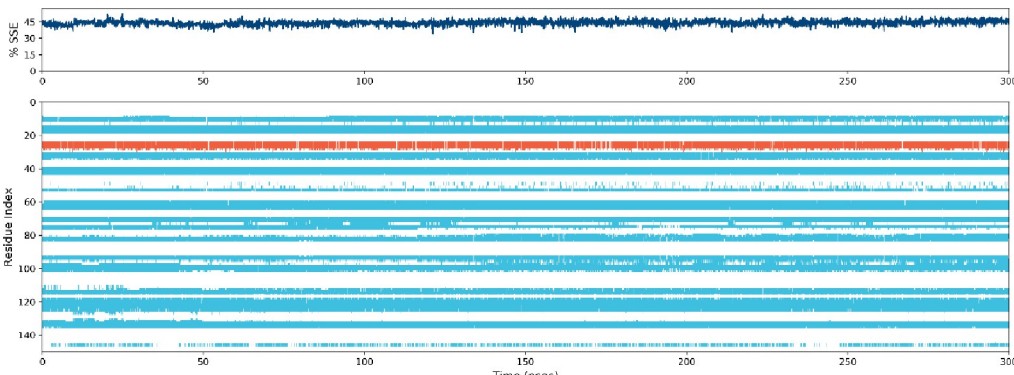

**Fig 6. Elements of protein secondary structure are dispersed across the protein-ligand complex over time of the simulation.** The alpha helices are represented by the red columns and the beta strands by the blue ones.

regions or N- and C-terminal regions (Fig 6). The constancy of ligand attachment to the protein is demonstrated by low RMSF estimates of attaching position residues. The secondary structure features of alpha-helices and beta-strands are tracked throughout the simulation (SSE). In the graph below, SSE is plotted against the residue index to display its distribution across the protein structure. Totaling 44.01 percent, it was found that helix made up 2.81 percent, and strand made up 41.20 percent of secondary structure elements.

In Fig 7, it is clear that hydrogen bonds and hydrophobic interactions constitute most of the important ligand-protein connections established by MD. Hydrogen bonds especially crucial for the amino acids were HIS_5, SER_5, and SER_87. For hydrophobic TYR_3, HIS_21, and TYR_90 are important. The ligand-protein interaction can be monitored over the course of the simulation. In the chart below the contacts and interactions are visualized in a timeline on this figure.

The MMGBSA method is frequently used to evaluate the binding energy of ligands to protein molecules. The influence of additional non-bonded interaction energies as well as the binding free energy of each 5280445_1z8r complex was evaluated. The binding energy of the

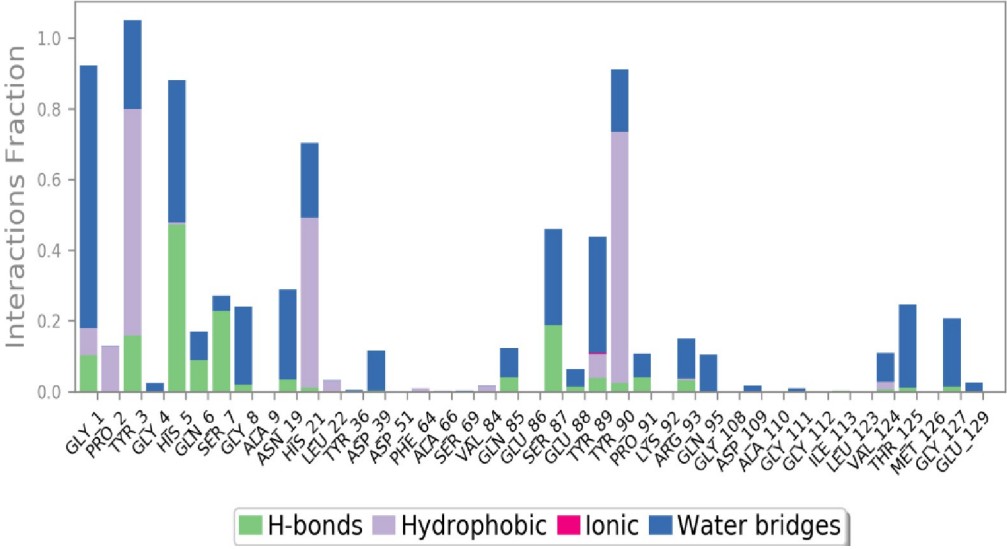

**Fig 7. Protein-ligand contact heatmap throughout trajectory (5280445_1z8r).**

**Table 2. Average MM-GBSA binding energy calculation of CID_5280445 with 1z8r after every 10 ns from MD simulation trajectories.**

| Energies (Kcal/mol) | 5280445_1z8r |
|---|---|
| $dG_{bind}$ | -45.5655 |
| $dG_{bind}Lipo$ | -7.6085 |
| $dG_{bind}vdW$ | -35.1296 |
| $dG_{bind}Coulomb$ | -14.6204 |
| $dG_{bind}H_{bond}$ | -1.3610 |
| $dG_{bind}Packing$ | -7.5653 |

ligand CID5280445 to 1z8r is -45.5655 kcal/mol. Gbind is governed by non-bonded interactions such as $G_{bind}Coulomb$, $G_{bind}Packing$, $G_{bind}H_{bond}$, $G_{bind}Lipo$, and $G_{bind}vdW$ (Table 2). The S2 Table contains all the MM-GBSA results. The average binding energy was mainly influenced by the $G_{bind}vdW$, $G_{bind}Lipo$, and $G_{bind}Coulomb$ energies across all types of interactions. The GbindSolvGB and Gbind Covalent energies, on the other hand, made the smallest contributions to the final average binding energies. Additionally, 5280445_1z8r complexes showed stable hydrogen bonds with amino acid residues by their $G_{bind}H_{bond}$ interaction values. As a result, the binding energy derived from the docking data was well justified by the MM-GBSA calculations that came from the MD simulation trajectories.

## Conclusion

Drug development has been researched extensively since the ability of transdisciplinary strategies to both speed up the process and reduce overall costs. The primary objective of this research was to discover target proteins for 2 Coxsackie B4 viral protein that causes T1D so that a lead drug could be selected for it. To counteract the effects of natural compounds on the 2 Coxsackie B4 viral protein, we chose substances that have this property. An appropriate natural inhibitor, identified from flavonoids CID: 5280445, blocks the action of 1z8r at its receptor. We reasoned that this material might act as a beginning point for the progress of a medication that targets Diabetes Type 1 (T1D) selectively without affecting other cellular processes. These results will be useful to researchers and may lead to the progress of a new medicine for the treatment of T1D.

## Supporting information

**S1 Table. Complete results of ADMET analysis.**
(CSV)

**S2 Table. MM-GBSA binding energy calculation of bonded and non-bonded interactions of CID_5280445 with 1z8r after every 10 ns from MD Simulation trajectories.**
(CSV)

**S1 Fig.**
(PNG)

## Author Contributions

**Data curation:** Wajeeha Rahman, Farhan Ullah, Anees Ullah, Muhammad Nasir Iqbal, Naveed Iqbal.

**Formal analysis:** Wajeeha Rahman, Anees Ullah.

**Methodology:** Zilong Zheng, Wajeeha Rahman, Naveed Iqbal.

**Supervision:** Shahid Ullah, Tianshun Gao.

**Validation:** Zilong Zheng, Wajeeha Rahman, Farhan Ullah, Anees Ullah, Muhammad Nasir Iqbal, Naveed Iqbal.

**Visualization:** Zilong Zheng, Wajeeha Rahman, Farhan Ullah, Anees Ullah, Muhammad Nasir Iqbal, Tianshun Gao.

**Writing – original draft:** Shahid Ullah.

**Writing – review & editing:** Shahid Ullah, Tianshun Gao.

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
