## [Decision Letter · Decision Letter 0]

28 Jun 2023

PONE-D-23-08995A computational approach to fighting type 1 diabetes by targeting 2C Coxsackie B virus protein with flavonoids.PLOS ONE

Dear Dr. Ullah,

Thank you for submitting your manuscript to PLOS ONE. After careful consideration, we feel that it has merit but does not fully meet PLOS ONE’s publication criteria as it currently stands. Therefore, we invite you to submit a revised version of the manuscript that addresses the points raised during the review process.

We look forward to receiving your revised manuscript.

Kind regards,

Erman Salih Istifli, PhD

Academic Editor

PLOS ONE

Journal Requirements:

A clean copy of the edited manuscript (uploaded as the new *manuscript* file).

“Dr. Shahid Ullah designed and supervised the project with Dr. Tianshun Gao's assistance and performed data analysis. Zilong Zheng, Farhan Ullah, Wajeeha Rahman, Dr. Anees Ullah contributed to data analysis. Shahid Ullah wrote the manuscript. All authors reviewed the manuscript.

This research is supported by National Natural Science Foundation of China [32100434] and Research Start-up Fund of the Seventh Affiliated Hospital, Sun Yat-sen University [ZSQYBRJH0020]”

Reviewers' comments:

Reviewer's Responses to Questions

**Comments to the Author**

1. Is the manuscript technically sound, and do the data support the conclusions?

Reviewer #1: No

Reviewer #2: Partly

2. Has the statistical analysis been performed appropriately and rigorously? 

Reviewer #1: N/A

Reviewer #2: No

3. Have the authors made all data underlying the findings in their manuscript fully available?

Reviewer #1: Yes

Reviewer #2: No

4. Is the manuscript presented in an intelligible fashion and written in standard English?

Reviewer #1: Yes

Reviewer #2: No

5. Review Comments to the Author

Reviewer #1: The article “A computational approach to fighting type 1 diabetes by targeting 2C Coxsackie B virus protein with flavonoids

This study will assist in developing medicine and testing it as a treatment for Diabetes Mellitus Type 1 brought on by Coxsackie B4 viruses by giving them an understanding of chemicals against these viruses. Although the study is valuable, it has some shortcomings.

Various situations should be considered that will increase the research value.

Abs tart should be rewritten with clear objectives and scientific language

Add the significance of your work.

Add limitation of the study

Typos should be corrected. The article should be accepted after Major revision.

Reviewer #2: The work is focusing on in silicon screening for a flavonoid inhibitor of a coxsackie B virus protein to combat type 1 diabetes (T1D).

The work is interesting but the preliminary nature of the results and lack of quantitative analysis make it much less interesting. Here are some specific points that the authors should address:

1. The writing could be much more concise and focused. The abstract should be more focus centered around the finding not the introduction about virus and T1D.

2. The material and method section could be significantly improved in that many detailed subsections could be merged into one concise experimental subsection for example, the first section about downloading coordinate from PDB and the preparation and evaluation of the structure for further docking analysis could be combined into “preparation of target structure”.

3. Also, the criteria why the authors did choose this protein in particular as a target should be further elaborated with references. Also, the authors should explain if there is any reason why “1Z8R” got chosen as a target?.

4. It should be also describe clearly what did the authors do to minimize and optimize the protein structure. I am very puzzling why did the author exploited computational server like SwissPDB to “optimize” the experimental crystal structure obtained from PDB?

5. The results from the binding site prediction to show all the putative active/allosteric sites should be included as a figure. The validation of the binding prediction is also important and should be added in the paper.

6. The ligand database is not clearly defined. It is unclear what computational or experimental approach the authors exploited to pre-screen the top 37 flavonoids before performing molecular docking with AutoDock Vina. The structures/structural illustration of all the compounds from the flavonoid library used in this work should be included as a supplementary material. Top 37 flavonoids with greatest binding should also be described structurally in the supplementary together with ADMET results.

7. Line 116 “Toxicity Analysis” is a misleading title for a subsection because in fact the toxicity analysis is just part of ADMET analysis. The authors should make it clear.

8. Lead identification section is also not very clear of how did the author choose lead. Is there any particular experiment or analysis the author perform to exclude potentially false hit out from the results apart from looking at the calculated binding energy and the drug-likeness result of ADMET analysis.

9. There is no need to add Ramachandran plot and the structure from PDB in the main figure. Figure 1 could be moved to supplementary information.

10. The structurally optimized model (perhaps superimposed with the PDB data) should be added in the figure and the pre-screening of the library should be at least mentioned. The figure to show binding prediction results should be added together with validation.

11. Figure 2 there is no detail about bond length in the figure. Also this pocket should be shown together with the full structure.

12.The Figure 3 that showed RMSD from simulations requires a lot of explanation. Is this one time simulation with no repeat? Normally, RMSD from all the repeat of MD simulation should be included in the figure. Also, is there any constraint used in the simulation? The authors should also discuss the fact that ligand RMSD shown in this figures could not displace beyond 1.5 A. Based on the pattern of the chart, it looks like there is a cutoff constraints at approximately 1.5 A?

13. The rest of the figures could be improved. Many of them should be included not as main figures but as supplementary figures.

14. All structures used in this study including protein, ligands, protein-ligand complexes should be deposited. The topology files used in trajectory analysis from MD simulations should be included as supplementary.

6. PLOS authors have the option to publish the peer review history of their article (what does this mean?). If published, this will include your full peer review and any attached files.

Reviewer #1: No

Reviewer #2: **Yes: **Puey Ounjai

---

## [Author Response · Author response to Decision Letter 0]

10 Aug 2023

Journal Requirements:

ANSWER: Addressed

A clean copy of the edited manuscript (uploaded as the new *manuscript* file).

ANSWER: Addressed. We copyedit the manuscript for language usage, spelling, and grammar.

“Dr. Shahid Ullah designed and supervised the project with Dr. Tianshun Gao's assistance and performed data analysis. Zilong Zheng, Farhan Ullah, Wajeeha Rahman, Dr. Anees Ullah contributed to data analysis. Shahid Ullah wrote the manuscript. All authors reviewed the manuscript.

This research is supported by National Natural Science Foundation of China [32100434] and Research Start-up Fund of the Seventh Affiliated Hospital, Sun Yat-sen University [ZSQYBRJH0020]”

ANSWER: This research is supported by National Natural Science Foundation of China [32100434] and Research Start-up Fund of the Seventh Affiliated Hospital, Sun Yat-sen University [ZSQYBRJH0020]”

Also The funders had the main role in study design, data collection and analysis, decision to publish, or preparation of the manuscript."

ANSWER: All the data is available in the main manuscript and in supplementary tables, and will be freely available under journal rule. 

ANSWER: Addressed

Reviewers' comments:

Reviewer's Responses to Questions

Comments to the Author

1. Is the manuscript technically sound, and do the data support the conclusions?

Reviewer #1: No

Reviewer #2: Partly

2. Has the statistical analysis been performed appropriately and rigorously?

Reviewer #1: N/A

Reviewer #2: No

3. Have the authors made all data underlying the findings in their manuscript fully available?

Reviewer #1: Yes

Reviewer #2: No

4. Is the manuscript presented in an intelligible fashion and written in standard English?

Reviewer #1: Yes

Reviewer #2: No

5. Review Comments to the Author

Reviewer #1: The article “A computational approach to fighting type 1 diabetes by targeting 2C Coxsackie B virus protein with flavonoids

This study will assist in developing medicine and testing it as a treatment for Diabetes Mellitus Type 1 brought on by Coxsackie B4 viruses by giving them an understanding of chemicals against these viruses. Although the study is valuable, it has some shortcomings.

Various situations should be considered that will increase the research value.

Abs tart should be rewritten with clear objectives and scientific language

Add the significance of your work.

Add limitation of the study

Typos should be corrected. The article should be accepted after Major revision.

ANSWER: Addressed. Comments addressed and manuscript improved appropriately. 

Reviewer #2: The work is focusing on in silicon screening for a flavonoid inhibitor of a coxsackie B virus protein to combat type 1 diabetes (T1D).

The work is interesting but the preliminary nature of the results and lack of quantitative analysis make it much less interesting. Here are some specific points that the authors should address:

1. The writing could be much more concise and focused. The abstract should be more focus centered around the finding not the introduction about virus and T1D.

ANSWER: Addressed.

2. The material and method section could be significantly improved in that many detailed subsections could be merged into one concise experimental subsection for example, the first section about downloading coordinate from PDB and the preparation and evaluation of the structure for further docking analysis could be combined into “preparation of target structure”.

ANSWER: Addressed.

3. Also, the criteria why the authors did choose this protein in particular as a target should be further elaborated with references. Also, the authors should explain if there is any reason why “1Z8R” got chosen as a target?.

ANSWER: Addressed.

4. It should be also describe clearly what did the authors do to minimize and optimize the protein structure. I am very puzzling why did the author exploited computational server like SwissPDB to “optimize” the experimental crystal structure obtained from PDB?

ANSWER: We minimized and optimized PDB structure because it has missing atoms and residues. Also it is crucial step for proper bond order in protein. 

5. The results from the binding site prediction to show all the putative active/allosteric sites should be included as a figure. The validation of the binding prediction is also important and should be added in the paper.

ANSWER: Addressed.

6. The ligand database is not clearly defined. It is unclear what computational or experimental approach the authors exploited to pre-screen the top 37 flavonoids before performing molecular docking with AutoDock Vina. The structures/structural illustration of all the compounds from the flavonoid library used in this work should be included as a supplementary material. Top 37 flavonoids with greatest binding should also be described structurally in the supplementary together with ADMET results.

ANSWER: ADMET analysis of compounds is provided in supplementary data. 37 flavonoid compounds were chosen from different literature reviews.

7. Line 116 “Toxicity Analysis” is a misleading title for a subsection because in fact the toxicity analysis is just part of ADMET analysis. The authors should make it clear.

ANSWER: Addressed.

8. Lead identification section is also not very clear of how did the author choose lead. Is there any particular experiment or analysis the author perform to exclude potentially false hit out from the results apart from looking at the calculated binding energy and the drug-likeness result of ADMET analysis.

ANSWER: Researchers employed a range of metrics to meticulously assess potential inhibitors, including docking scores, interactions, logP coefficient, rotatable bonds, ring presence, PSA, Blood-Brain Barrier permeability, and Ames Toxicity. This multi-dimensional approach narrowed down the pool of inhibitors to a promising subset. Notably, the 2C Coxsackie B virus protein inhibitor stood out due to low binding affinities and strong lead-likeness scores. Positive interactions and compatibility with the target protein further supported its selection. The thorough use of diverse metrics led to identifying the 2C Coxsackie B virus protein inhibitor as a high-potential option for drug development.

9. There is no need to add Ramachandran plot and the structure from PDB in the main figure. Figure 1 could be moved to supplementary information.

ANSWER: Addressed.

10. The structurally optimized model (perhaps superimposed with the PDB data) should be added in the figure and the pre-screening of the library should be at least mentioned. The figure to show binding prediction results should be added together with validation.

ANSWER: Addressed in fig 1.

11. Figure 2 there is no detail about bond length in the figure. Also this pocket should be shown together with the full structure.

ANSWER: Addressed

12.The Figure 3 that showed RMSD from simulations requires a lot of explanation. Is this one time simulation with no repeat? Normally, RMSD from all the repeat of MD simulation should be included in the figure. Also, is there any constraint used in the simulation? The authors should also discuss the fact that ligand RMSD shown in this figures could not displace beyond 1.5 A. Based on the pattern of the chart, it looks like there is a cutoff constraints at approximately 1.5 A?

ANSWER: We used state of the art tool to perform a very long 300 ns simulation. All the parameters are mentioned in the manuscript. We did not use any cutoff to limit the RMSD of the ligand. This behavior of ligand RMSD is explained in the manuscript. For further investigation, trajectory files could be found at this link: https://drive.google.com/file/d/1hpx9TGYe2nA4z9VOhvwgrvsoBV7Y8UEM/view?usp=sharing. 

13. The rest of the figures could be improved. Many of them should be included not as main figures but as supplementary figures.

ANSWER: Addressed

14. All structures used in this study including protein, ligands, protein-ligand complexes should be deposited. The topology files used in trajectory analysis from MD simulations should be included as supplementary.

ANSWER: Addressed

---

## [Editor Report · Decision Letter 1]

11 Aug 2023

A computational approach to fighting type 1 diabetes by targeting 2C Coxsackie B virus protein with flavonoids.

PONE-D-23-08995R1

Dear Dr. Ullah,

We’re pleased to inform you that your manuscript has been judged scientifically suitable for publication and will be formally accepted for publication once it meets all outstanding technical requirements.

Kind regards,

Erman Salih Istifli, PhD

Academic Editor

PLOS ONE
---

## [Editor Report · Acceptance letter]

18 Aug 2023

PONE-D-23-08995R1 

A computational approach to fighting type 1 diabetes by targeting 2C Coxsackie B virus protein with flavonoids. 

Dear Dr. Ullah:

I'm pleased to inform you that your manuscript has been deemed suitable for publication in PLOS ONE. Congratulations! Your manuscript is now with our production department. 

Kind regards, 

on behalf of

Assoc. Prof. Dr. Erman Salih Istifli 

Academic Editor

PLOS ONE